# High-Speed Tracking with Mutual Assistance of Feature Filters and Detectors

**DOI:** 10.3390/s23167082

**Published:** 2023-08-10

**Authors:** Akira Matsuo, Yuji Yamakawa

**Affiliations:** 1Graduate School of Interdisciplinary Information Studies, The University of Tokyo, Tokyo 153-8505, Japan; 2Interfaculty Initiative in Information Studies, The University of Tokyo, Tokyo 153-8505, Japan; y-ymkw@iis.u-tokyo.ac.jp

**Keywords:** high-speed vision, image processing, machine learning, object tracking

## Abstract

Object detection and tracking in camera images is a fundamental technology for computer vision and is used in various applications. In particular, object tracking using high-speed cameras is expected to be applied to real-time control in robotics. Therefore, it is required to increase tracking speed and detection accuracy. Currently, however, it is difficult to achieve both of those things simultaneously. In this paper, we propose a tracking method that combines multiple methods: correlation filter-based object tracking, deep learning-based object detection, and motion detection with background subtraction. The algorithms work in parallel and assist each other’s processing to improve the overall performance of the system. We named it the “Mutual Assist tracker of feature Filters and Detectors (MAFiD method)”. This method aims to achieve both high-speed tracking of moving objects and high detection accuracy. Experiments were conducted to verify the detection performance and processing speed by tracking a transparent capsule moving at high speed. The results show that the tracking speed was 618 frames per second (FPS) and the accuracy was 86% for Intersection over Union (IoU). The detection latency was 3.48 ms. These experimental scores are higher than those of conventional methods, indicating that the MAFiD method achieved fast object tracking while maintaining high detection performance. This proposal will contribute to the improvement of object-tracking technology.

## 1. Introduction

Object tracking using camera images is an essential technology for computer vision and has been applied to various systems, such as automatic driving and robotics. In the field of robot manipulation, for example, highly automated manipulators have been developed by integrating robot control and vision systems. These robots use cameras for object recognition and localization, which has improved the versatility of manipulation [1]. On the other hand, dynamic manipulation using a high-speed camera has also been studied in recent years [2]. These highly responsive machine vision systems also contribute to the development of industrial and service robots in terms of improved work efficiency. Therefore, tracking systems require advanced image recognition and high operating speeds to detect fast-moving objects quickly.

Many tracking algorithms have been developed in previous studies. One of the most common methods is image feature matching. This method extracts features from each camera frame to find the target object. Various types of features, such as shape, color, and contour, are selected depending on the target object [3]. In addition, more generalized local image features such as Histograms of Oriented Gradients (HOG) [4] and the Scaled Invariance Feature Transform (SIFT) [5] are also often used. Another approach is region-based tracking, where a bounding box is used to create a template of the image around the object, and each frame is matched to locate the target. In particular, tracking using correlation filters is known to be fast and is widely used in object-tracking research [6]. However, while these region-based trackers with correlation filters have high processing speeds, they are poor at long-term tracking because many of these methods have relatively low noise tolerance and cannot cope with scale or aspect changes of the bounding box.

On the other hand, deep learning techniques are effective for object detection, and many studies have been conducted [7]. Deep learning-based detection has made it possible to find objects from complex images with high accuracy. There is a way to detect objects frame by frame in the camera by using a pre-trained model. However, due to computational costs, detection methods based on deep learning have lower tracking speeds than feature matching methods.

From the above, each of the conventional methods has its advantages and disadvantages. Generally, tracking methods using image feature extraction and template matching consist of relatively simple algorithms. Region-based object tracking, mainly represented by correlation filters, enables high-frame-rate processing by optimizing matching calculations. However, they can only be used in restricted environments because they track the visibility of objects in a specific background. In contrast, deep learning-based tracking is effective in any situation due to its high detection performance. Meanwhile, processing delays make it difficult to track fast objects and reduce responsiveness. To summarize, there is a trade-off between processing speed and detection performance.

The characteristics of conventional methods can be illustrated in Figure 1. Figure 1 shows that there is a trade-off between processing speed and detection performance.

Therefore, this work aims to resolve this trade-off to achieve fast and stable tracking. For this purpose, we propose a tracking system that combines multiple types of methods. Although algorithms using several trackers already exist, in this study, we build a system in which trackers assist each other to improve overall performance. Our motivation can be represented graphically as shown in Figure 1. Our goal is to contribute to the development of single-object-tracking technology through the proposed method, named “Mutual Assist tracker of feature Filters and Detectors (MAFiD method)”.

## 2. Related Work

Tracking algorithms can be classified into two types: object detection and inter-frame tracking. Here, we present some previous work on each of these methods.

### 2.1. Object Detection

#### 2.1.1. Manual Feature Extraction

Various methods have been proposed to extract features from images for target detection. A simple example is thresholding to a specific brightness or color in a frame image, which can easily detect bright objects like optical markers. Other methods also focus on local patterns of light and dark. Haar-like features can be combined with simple geometric patterns to represent the contour of the object [8]. Harris et al. proposed a method to detect edges and corners by extracting the gradient of pixel values and performing principal component analysis [9]. However, these methods are highly dependent on the background and lighting environment, which reduces their tracking accuracy. As a method for extracting more generalized features, HOG [4] and SIFT [5] create histograms of the local gradients in the image to obtain more complex features. SIFT is more robust to rotation and scale changes but is computationally expensive.

#### 2.1.2. Motion Detection

Motion detection is also effective for detecting moving objects. For instance, frame subtraction is a simple method of finding the motion by comparing several consecutive frames. Background subtraction finds the difference between a current frame and a background image [10]. However, the subtraction method has the disadvantage of being susceptible to noise, like flickering. Thus, it is necessary to combine morphological transformations and smoothing filters or to keep updating the background image online. Optical flow is a method for estimating the direction of object movements based on brightness changes in the image. In particular, the Lucas–Kanade method is widely used for estimation [11]. Optical flow can calculate a vector map of object motion; however, it also has limitations in some environments where objects move at high speed or change their appearance.

#### 2.1.3. Deep Learning-Based Detection

With the development of machine learning technology, there has been much research on object detection using deep learning. Deep learning-based detection is more accurate because it can learn to extract features that were previously extracted manually. Detection algorithms are classified into one-stage and two-stage types [12]. Two-stage detectors learn separate models for object region detection and object recognition. A typical example is the Region-based Convolutional Neural Network (R-CNN) [13]. Advanced models such as Fast R-CNN [14] and Faster R-CNN [15] have also been developed. In particular, Faster R-CNN achieves higher processing speed by introducing a Region Proposal Network (RPN) that efficiently searches for candidate object regions. Feature Pyramid Network (FPN) is another method that enhances multi-scale detection performance [16].

One-stage detectors have a single model that estimates bounding boxes and classification. In 2016, You Only Look Once (YOLO) was released, improving processing speed over the two-stage detectors [17]. YOLO is still under development and has been released in its latest models [18]. It divides the image into many grids and performs object detection and classification in each grid. While YOLO is robust to background detection errors, it is less accurate when objects appear small or dense. Unlike YOLO, a Single Shot Multi-box Detector (SSD) constructs feature maps from multiple layers in neural networks, enabling higher accurate detection in dense environments and various sizes [19].

Since deep learning-based detection can extract more complex features than manual filters, it can achieve stable performance in any environment. In addition, one-stage methods have recently become fast enough to be used in real time. However, the processing speed is still insufficient to perform fast-tracking with a high-speed camera in this study.

### 2.2. Inter-Frame Tracking

Object detection finds objects in every frame, whereas inter-frame tracking locates moving objects in consecutive video frames. The tracking task is to extract the feature from each frame around the object region and to detect the same pattern in the next frame. Inter-frame tracking algorithms generally do not have object recognition or classification functions. Instead, they tend to focus on real-time performance and often require robustness to track objects even if their visibility changes.

#### 2.2.1. Template Matching

Template matching is known to be a relatively accurate and fast region-based tracking method. It identifies object locations by calculating the similarity between a pre-defined template image and the current frame image. While there are several ways to obtain templates, the object tracker uses the update template method in which an internal template image is updated sequentially. In particular, the matching method using a correlation filter is known to be relatively fast [6]. Correlation filters use fast Fourier transforms to reduce the computational cost of creating and updating template images. There are many algorithms for region-based template tracking using correlation filters.

Here is a description of the process steps for tracking using a correlation filter. First, the tracker initializes the correlation filter in advance with the first frame and the target bounding box. This initialization determines parameters such as filter size and aspect ratio. In the tracking phase, features are extracted from the camera frame, and a cosine window is applied to remove the influence of boundary areas. Next, a correlation filter is applied to obtain a response map. The tracker can estimate the direction and distance of the object’s motion from the response map to output the bounding box. At the same time, a desired response map is created based on the current object position. The final process involves updating the internal correlation filter by applying it to the desired response map. The above sequence of steps is performed frame-by-frame, which enables real-time tracking while updating the filter online.

There are various types of tracking methods using correlation filters, depending on the feature extraction method. The method in which pixel values are directly input as feature values is known as the Minimum Output Sum of Squared Error (MOSSE), which achieves high-speed tracking at over 500 frames per second (FPS) [20]. A spatiotemporal context (STC) learning tracker utilizes time-series information by providing multiple past frames as inputs [21]. The Kernelized Correlation Filter (KCF) extended the method to nonlinear systems using kernel tricks [22]. KCF optimizes computational costs by introducing a cyclic matrix and discrete Fourier transform. In addition, it supports multichannel color as well as gray-scale images. An extended KCF model that adds HOG features to improve accuracy has also been reported. More recent models based on correlation filters focus on the object size and attempt to vary the scale and aspect ratio of the bounding box. Discriminative Scale Space Tracking (DSST), for example, trains internal filters with multiple-scale images [23]. Another method [24] adds a scale estimation function to the previous tracking structure. On the other hand, MUlti-Store Tracker (MUSTer) retains multiple memories from short to long term, making it robust to noise caused by temporal changes in object visibility [25].

#### 2.2.2. Deep Learning-Based Tracking

There are also studies on object tracking using deep learning technology. Generic Object Tracking Using Regression Networks (GOTURN) uses a pre-trained model [26]. Unlike object detection, GOTURN does not perform classification and only determines whether the object is a target or not. In the tracking process, two consecutive frames are input to predict the change in the object’s position. Multi-Domain Convolutional Neural Network (MDNet) combines offline training and online training [27]. The network model of MDNet consists of a shared layer and a domain layer. Before tracking, multi-domain learning is performed on several video datasets, which allows the shared layer to learn domain-independent features. During tracking, MDNet updates the bounding box while training the shared layer and the newly initialized domain layer.

Tracking methods combined with a Siamese network have also been proposed. SiamFC (Siamese Fully-Convolutional Network) has a network for deep metric learning [28]. Its convolutional layers extract features from two input images—a template image and the current frame image—to obtain a similarity map. The map is used to locate the object in the frame image. In addition, various advanced algorithms have been developed based on the basic structure of SiamFC. For instance, a method that combines a Sham network and a correlation filter has been proposed to improve the learning efficiency of similarity maps [29]. The SiamRPN (Siamese Region Proposal Network) tracker enhances tracking accuracy and speed [30]. The SiamRPN model includes the Siamese network and the RPN structure used in Faster R-CNN. RPN detects object regions based on the features extracted by the Siamese network, which enhances the detection speed of the SiamRPN. More recently, new models have been proposed, such as SiamRPN++ with ResNet [31] and SiamFC++, which is robust to rapid changes in visibility [32].

### 2.3. Hybrid Method

Some research has also been conducted to improve tracking performance by combining multiple methods. Ma et al. attempted to perform efficient feature extraction by using some layers of a CNN [33]. They analyzed the convolutional layers of the network model for object recognition and found that some layers can extract edges that are effective for target detection. They achieved tracking by using some of the layers to acquire image features and inputting them into a correlation filter. Although this method is computationally more expensive than correlation filter-based tracking, it achieves stable and long-term tracking.

Tracking-by-Detection (TBD) is a widely used method that integrates object detection and object tracking. Inter-frame object tracking is called Model-Free-Tracking (MFT) because it does not perform object recognition but instead tracks the same object in the previous frame. In contrast, TBD uses a detector when finding or re-detecting an object [34]. Therefore, TBD enables longer-term tracking. Liu et al. proposed a TBD method that includes fast DSST (fDSST) for object tracking and SSD for object detection [35]. While fDSST tracks an object in every frame, SSD detects an object at certain intervals. The correlation filter is reconstructed based on the detection results, including size and aspect ratio changes. Wang et al. proposed a method that combines tracking using HOG features and multiple correlation filters with YOLOv3 detection [36]. They also introduced an Average Peak-to-Correlation Energy (APCE) score to evaluate the time variation of the response map in the correlation filter. YOLOv3 re-detects when the APCE score is low, enabling automatic re-tracking even if tracking fails due to occlusion or noise.

Although the above method improves the tracking stability, it has the disadvantage of lowering the processing speed because of interruption of the detection process. To solve this problem, Jiang et al. proposed running a correlation filter-based tracker and a deep learning-based detector simultaneously with parallel processing [37]. The detection results constantly update the tracker, which allows periodic updating of the correlation filter without the need to determine tracking failures. In addition, parallel processing ensures that the tracker and the detector are not affected by each other’s processing speed. Nevertheless, despite these innovations, the bottleneck in the overall system has not been solved yet because the re-tracking speed depends on the detector’s latency.

### 2.4. Challenges of Previous Studies

While inter-frame tracking is relatively fast, many of these methods do not have a detection function and are designed for short-term tracking. Object detection, on the other hand, can locate the position of a target object frame by frame, although it requires a longer processing delay. Hybrid methods represented by TBD achieve occlusion-resistant tracking by combining multiple algorithms, but bottlenecks remain in terms of processing speed. These conventional methods focus on either short-term or long-term tracking, leaving the trade-off between accuracy and speed unresolved. As a result, they have performance limitations that depend on the tracking environment. To resolve this trade-off, it is not sufficient to simply put multiple methods together. Future tracking models need a mechanism to overcome bottlenecks and improve system performance through the synergistic effects of each method. Therefore, this study aims to construct an algorithm that effectively integrates tracking and detection techniques. In this paper, we attempt to improve the overall processing speed and accuracy by optimizing both the filter updating and the object detection process.

## 3. Proposed Methodology

This work proposes an object-tracking method with a high frame rate and high detection performance. To achieve this, we aim to increase the speed of both the tracking algorithm and detection processes. In this section, we provide an overview of the proposed MAFiD method and explain the features and advantages of each tracking and detection process in comparison with conventional methods.

### 3.1. Overview of the Tracking Method

The previous section introduced various tracking methods, each of which has advantages and disadvantages in terms of speed and accuracy. The proposed MAFiD method combines several trackers and detectors to compensate for the disadvantages of each algorithm.

The MAFiD method integrates three methods: inter-frame tracking using correlation filters, object detection using deep learning, and motion detection using background subtraction. The correlation filter has a low computational cost for high-speed tracking. However, because the tracker looks at the difference between frames, it may lose sight of the object due to changes in the object’s visibility. Moreover, it cannot change the size and aspect ratio of the target region. As a result, the tracking position may gradually shift as the processing time increases, which means that the inter-frame tracker is not suitable for long-term tracking. In order to sustain stable tracking for a long time, the MAFiD method adopts deep learning. Additionally, it also uses motion detection to locate new objects in a relatively short time.

An overview of the proposed MAFiD method is shown in Figure 2. The role of the entire system is to acquire a camera image and output a Region of Interest (ROI) where the target object exists in the form of a rectangular area. As shown in Figure 2, Correlation Filter (CF)-based tracking and Deep Learning (DL)-based detection run in parallel. Because high-speed cameras capture images at high frame rates, the distance an object moves between each frame is expected to be very short. The MAFiD method constantly outputs the results of CF-based tracking as a short-term tracker. In the detection process with deep learning, it is more efficient to search only around the object position in the previous frame than to search the entire image. Thus, the MAFiD method crops images of areas near the ROI tracked in the previous frame before object detection. This contributes to reducing the search area and computational cost. After detection, the ROI is used to initialize the correlation filter to recover from a tracking failure or to correct the target position. Motion detection is used when an object is first found or when the tracker loses the object. The motion detector calculates the center position of the object and modifies the ROI position of the correlation filter. Re-detection by the motion detector is only done when an object is missing, such as when the CF-based tracker fails to update the filter or when the DL-based detector cannot find the target object.

Figure 3 shows a flowchart of the proposed MAFiD system. The process consists of two threads running in parallel. These threads are communicating asynchronously with each other. Thread 1 first acquires frame images from the camera and updates the correlation filter. It modifies the current correlation filter using detection results obtained from thread 2. If the filter has not been successfully updated for some time, the system assumes that tracking and detection have failed. In that case, motion detection is performed to fix the ROI position. By executing this loop process at high speed, tracking results are output with low latency and at high frequency. In Thread 2, a search area is cropped from the frame image by using the ROI output from Thread 1. After detecting the target object in the cropped image, the system creates the new tracker, initializes it with the detected area, and sends a new correlation filter to Thread 1. The unique feature of this method is that the CF-based tracking and DL-based detection are not completely separated but rather take advantage of each other’s outputs. This can achieve higher accuracy and speed than simply using a tracker and detector separately in parallel.

As shown in Figure 2, the proposed MAFiD method consists of two functions: the tracking part, which tracks objects from frame to frame, and the detection part, which finds objects first. Section 3.2 and Section 3.3 below describe the tracking and detection mechanisms in detail, respectively.

### 3.2. Tracking Process

Here, we describe the details and advantages of the MAFiD method in terms of the tracking process. For algorithm comparison, the TBD method is chosen as the conventional method. TBD is a hybrid method combining a tracker and a detector, similar to the MAFiD method presented in the previous section. A comparison of the object-tracking process between the conventional TBD and the proposed MAFiD method is shown in Figure 4.

After locating the object with a DL-based detector, TBD tracks the object with a CF-based tracker using the ROI obtained from the detection results. The object position detected in the previous frame is applied to the latest frame image to create a new correlation filter. Although the TBD method is effective when the object speed is relatively slow, it is not suitable for fast-moving objects. That is because object detection by deep learning takes a relatively long time. During detection, the object is moving before the correlation filter is created. As a result, the correlation filter is initialized while the object position is shifted or the ROI region is inaccurate. In contrast, the proposed MAFiD method does not apply the detected ROI directly to the latest frame image. Instead, it applies a new correlation filter created from the detection results to the latest frame. This mechanism solves the problem of ROI and filter initialization errors due to delays in the detection process. In addition, since the CF-based tracker and the DL-based detector run in parallel, the system can always output ROIs at high frequency. In consequence, the proposed MAFiD method can continue tracking while maintaining high matching accuracy, whereas the conventional TBD method causes filter-matching errors during high-speed tracking.

This study also proposes a mechanism to update the correlation filter based on the detection results. In frame t5 in Figure 4, the system has two filters: the correlation filter used for tracking in frame t4 and the correlation filter created with the ROI detected from frame t1. Figure 5 shows the process flow of updating these filters and applying them to frame t5.

Since the object tracker is designed to track a target area in a series of frames, it searches for the current object position based on where the object has existed in previous frames. In particular, CF-based tracking methods restrict the search area so that the matching sensitivity is high near the ROI of the previous frame [6]. For this reason, they maintain both the filter and the search area during the tracking process. In Figure 4, if the correlation filter created by the detection in frame t1 is applied at frame t5, the object position in frame t5 is far from the tracker’s search area. That is to say, the delay in DL-based detection from frame t1 results in a large gap between frames t1 and t5.

To solve this problem, the proposed MAFiD method adjusts the search area during the filter switch process, as shown in Figure 5. It refers to the tracking results in frame t4 to obtain an effective search area. Since the system replaces only the filter from the tracker running in parallel at frame t4, it can continue tracking without losing the object, even if there is a delay in the detection process. Moreover, when updating the filter at frame t5, the MAFiD method does not always use the result of DL-based detection. This is because filters obtained from object detection often cause matching failures in later tracking. If the target texture dynamically changes due to object rotation or movement, the object’s appearance may change significantly from object detection to filter updating. In such a situation, even if the correlation filter is created after the object is successfully detected, the filter update is likely to fail in the latest frame. Considering such a case, the MAFiD method selects an appropriate correlation filter between the one created by the detection at frame t1 and the one used at frame t4. After applying each of the two correlation filters to frame t5, the MAFiD method compares their reliability and adopts the one with higher matching accuracy.

To select one of the two correlation filters, the response map of each tracker is used for comparison. The response map is the output image obtained by applying a correlation filter to the frame image. The tracker estimates the position with the highest value on the map as a candidate for the next object position. By looking at the map, we can evaluate the reliability of the matching. The proposed MAFiD method uses the Peak-to-Sidelobe Ratio (PSR) to calculate the matching accuracy. The PSR value in the response map bmx is calculated by: (1)$$\mathrm{PSR}\left(x\right)=\frac{\max(x)-\mathrm{mean}(x)}{\mathrm{std}(x)}.$$

According to Figure 5, the MAFiD method uses each correlation filter and the frame t5 image to create response maps and calculates their PSR values. By comparing those PSR values, a correlation filter with better matching accuracy can be selected.

### 3.3. Detection Process

Next, we describe the features of the proposed MAFiD method in terms of the detection process. Similar to the previous explanation of the tracking process, we will take TBD as a conventional hybrid method and compare it with the MAFiD method. Figure 6 shows the process flow from the appearance of an object in the camera frame until tracking is performed by the correlation filter. Since inter-frame tracking methods cannot identify the object region from the entire image, hybrid trackers must be used in combination with other object detection methods. In the TBD method, the ROI of the object region is first obtained with the DL-based detector, and the tracker is then switched to the CF-based tracker in subsequent frames. However, because the high computational cost of object detection causes a delay, it takes a long time to start tracking. Furthermore, the detection delay may also cause matching failures and ROI position errors, as in Figure 4. In consequence, conventional methods are not suitable for situations where objects are moving at high speed.

The proposed MAFiD method reduces processing costs by introducing motion detection of the background subtraction to estimate object positions at high speed. However, while background subtraction is computationally inexpensive, it has some disadvantages. It is vulnerable to noise caused by the lighting environment and cannot accurately recognize object contours. Hence, the MAFiD method obtains the center of the background subtraction area and roughly estimates the target location. Even if the size and aspect ratio of the ROI does not exactly match the current object area, it is possible to track with the correlation filter relying on the rough object location. This allows the tracker to detect a target with low latency and start tracking it immediately after it appears in the frame. Motion detection is only a means of predicting candidate object locations and does not necessarily detect objects accurately in every frame. Therefore, the MAFiD method runs the motion detector until the correlation filter succeeds in tracking and stops it during the tracking process shown in Figure 4. We refer to the PSR value calculated by Equation (Equation 1) to determine if the CF-based tracking is successful or not. In other words, we activate the motion detection only when the PSR value is lower than a certain threshold value.

### 3.4. Algorithms

Here, we describe the algorithms for the tracker and detector. Since fast-tracking is essential in this study, we use the MOSSE tracker as the CF-based tracker. MOSSE is one of the fastest tracking algorithms with a correlation filter. It has the advantage of low computational costs because it directly inputs each pixel value as an image feature. Therefore, the proposed MAFiD method utilizes MOSSE for short-term tracking because it can track fast objects with low latency. For the same reason, we use YOLOv5 as a DL-based detector from a high-speed processing viewpoint [38]. YOLO is a one-stage detection method and is known to be as accurate as and faster than other learning-based detection methods. We also use a dynamic background subtraction method for motion detection. While the background subtraction method has lower detection accuracy, it is much less computationally expensive and faster than DL-based detectors to find object locations. The motion detector retains the background image dynamically. Using the *n*-th frame image (I(n)), the internal background image B(n) is given by:(2)B(n)=αI(n)+(1−α)B(n−1).

As shown in Equation (Equation 2), the dynamic background image (B(n)) is the accumulated and smoothed image of the past frames. The variable α means the contribution of the current frame to the background image. When performing the detection process, the motion detector calculates the absolute difference between the current frame (I(n)) and the internal background image (B(n−1)). Finally, the motion detector can find the moving object’s position by applying a threshold to the resulting image.

This section introduces a method for tracking fast-moving objects by combining multiple trackers and detectors. The novelty of the proposed MAFiD method can be mainly summarized in three points. First, the CF-based tracker and the DL-based detector run in parallel and assist each other to achieve fast and accurate tracking. Second, the results of tracking and deep learning detection are compared in terms of their PSR values to select the appropriate correlation filter. Third, the motion detection and CF-based tracker are used to reduce the delay of the re-detection process.

## 4. Experiments and Discussion

In this chapter, the performance of the proposed MAFiD method and conventional methods is evaluated through experiments. They need to be measured from two perspectives: tracking and detection. The performance of frame-to-frame matching is important for tracking and the delay time is important for detection. Therefore, in this chapter, we prepare the experiments in Section 4.1 and Section 4.2 and conduct tracking and detection experiments in Section 4.3 and Section 4.4, respectively.

### 4.1. Experimental System

In this section, we describe the experimental system for tracking. Figure 7a shows the camera device for capturing images of objects. A high-speed camera (MQ013MG-ON, Ximea Corp., Lakewood, CO, USA) was used with a ring-shaped white LED (WDR-FH223) as a light source. The camera captured images in one channel (grayscale image) with a resolution of 544 × 438, an exposure time of 1.4 ms, and a lens focal length of 6 mm. The distance between the camera and the target object was about 1 m. The computer for image processing was connected to the camera via a USB 3.1 Gen1 interface and had the following hardware specifications: Intel Core i9-10980XE, 3.00 GHz CPU, 128 GB memory, and NVIDIA RTX A5000 GPU (24 GB VRAM).

We used a transparent plastic capsule shown in Figure 7b as the target object for the tracking experiment. Since one of the goals of this work is to demonstrate the detection performance of the MAFiD method, it is necessary to show that the MAFiD method can detect targets that are difficult to detect with conventional inter-frame trackers. Therefore, the target object must not have a specific texture or brightness pattern that changes its appearance frequently so that it cannot be detected easily by pattern matching. For this reason, we used a semi-transparent object that changed its texture, depending on the background environment. Since a semi-transparent object is difficult to track with high-speed methods such as template matching, advanced detection using machine learning is required to achieve long-term tracking. The use of the semi-transparent object in Figure 7b means that this experiment was conducted under a more complex environment than previous studies of high-speed tracking. Through this experiment, we evaluated the improvement in tracking performance in this work.

### 4.2. Training of DL-Based Detector Model

Since the proposed MAFiD method includes object detection with YOLOv5, the learning model has to be trained in advance. First, we used the high-speed camera in Figure 7a to capture the plastic capsule in Figure 7b and prepared 1760 grayscale images of 554 × 438 pixels. These images were rotated, flipped, and changed in brightness to expand the number of images to 8800. We prepared the labeled data by making a rectangle of the object region in each image. As in Figure 2, the input image for YOLO detection was cropped from the camera frame in the MAFiD method. Thus, it was necessary to crop the training images around the object as in the actual use. The differences in the procedure for creating a dataset between the regular YOLO and the YOLO of the MAFiD method are shown in Figure 8.

The MAFiD method can suppress the search cost of detection by inputting images only around the ROI of the tracking results. In the same way, the cropped image was simulated in the training phase by randomly clipping the area around the object from the dataset images. However, since the cropped image can be of various sizes, the input size was fixed by resizing it to 128 pixels. As a result, the MAFiD method could detect the object with a smaller search area. The crop operation also prevented the resolution of the input image from being reduced after resizing.

YOLOv5 was trained on 8000 images as training data and the remaining 800 images as cross-validation data. The training results for 200 epochs on the two types of datasets in Figure 8 are summarized in Table 1.

In Table 1, the mean Average Precision (mAP) is an evaluation index that averages the area of the Precision–Recall Curve (AP value), which represents the relationship between the precision and the recall value. A higher mAP value means higher accuracy in machine learning. The mAP(0.5–0.95) is the average AP value when the threshold for the ROI overlap area is 0.5–0.9. Consequently, Table 1 shows that the YOLO model of the MAFiD method maintained high detection scores even for small input sizes.

These indices used Intersection over Union (IoU) to evaluate the accuracy of the ROI. Given two object regions, ${\mathrm{ROI}}_{A}$ and ${\mathrm{ROI}}_{B}$, IoU was calculated as in Equation (Equation 3). An IoU close to 1 means that the ROI region was highly accurate.
(3)$$\mathrm{IoU}=\frac{Intersection}{Union}=\frac{Area({\mathrm{ROI}}_{A}\cap {\mathrm{ROI}}_{B})}{Area({\mathrm{ROI}}_{A}\cup {\mathrm{ROI}}_{B})}$$

### 4.3. Tracking Experiment

This experiment mainly evaluated the speed of inter-frame tracking. A thin string was tied to the plastic capsule that was the target object. During the experiment, the capsule was rotated by swinging it at high speed in the air. The object was captured by a high-speed camera and tracked by the proposed MAFiD method and conventional methods. To compare each tracker using the same frame image, the tracking process was performed offline after taking the video in advance. The xy coordinates were set in the horizontal and vertical directions of the camera image.

The tracking methods to be compared in this experiment are listed below.

Proposed MAFiD MethodMOSSE tracker + YOLOv5 detector + Motion detectorCF-based tracking methodMOSSE trackerDL-based detection methodYOLOv5 detectorControl methodMOSSE tracker + YOLOv5 detector (Global detection)

We chose the MOSSE tracker, a fast method using correlation filters, and the YOLOv5 detector, a method using deep learning, as conventional methods. They were compared with the MAFiD method to verify the tracking speed and accuracy. Since the MOSSE tracker did not have a detection function, the ROI of the first frame was specified manually in advance. In addition, to investigate the effect of the local detection described in Figure 4, we also added a control method that does not crop the frame image (YOLOv5 searches the entire image).

The ROI during tracking using each method is shown in Figure 9.

As the experimental environment, we set up a background with large differences in lightness and darkness, as shown in Figure 9. When a semi-transparent object like a plastic capsule moves in this environment, the appearance of the object changes drastically depending on the brightness of the background. Matching methods using correlation filters are vulnerable to this drastic change because they deal with correlation between frames. According to Figure 9, the MOSSE tracker failed to track just when the background brightness changed and lost sight of the object. In contrast, YOLO performed detection at every frame, which prevented it from losing sight of the object, though the delay in processing time caused the misplacement of the ROI from its actual position. Meanwhile, the MAFiD method had a small ROI error due to delay and achieved stable tracking without losing sight of the object.

The trajectory of the x-coordinate of the center position of the ROI for each method is plotted in Figure 10. ROIs were also created manually for every frame image in the video to plot the ground truth line. Figure 10 shows that MOSSE tracking failed to update the ROI in the process, indicating that it could not track for a long time. Compared to the MAFiD method and MOSSE tracking, object detection by YOLO had a delay from the ground truth due to the higher processing cost. It can also be seen that the YOLO graph has a staircase shape. This is because YOLO could not keep up with the frame rate of the high-speed camera due to the longer detection time.

We evaluated the accuracy of the ROI region in Figure 11. The IoU shown in Equation (Equation 3) was used to quantify the overlap rate of the ROI relative to the ground truth. Looking at the YOLO graph, we can find that the IoU values oscillated. This is because the object moved even when the ROI was not updated during the YOLO process. It also shows that the accuracy of the IoU values went up and down periodically. This means that the tracking accuracy of the DL-based detector decreased as the object moved faster.

On the other hand, focusing on the MAFiD method and the control method, both can track with relatively high accuracy. However, we can confirm that the accuracy of the control method temporarily decreased when the background brightness or darkness changed significantly. In this case, it is inferred that the MOSSE tracker lost the object and waited for the detection result by YOLO. Since the control method used YOLO to search from the entire image, it took a relatively long time to resume tracking. In contrast, the MAFiD method locally searched only around the location where MOSSE was last tracked. Thus, it enabled faster detection. The MAFiD method succeeded in reducing the recovery time after a target was lost by efficiently taking advantage of both MOSSE and YOLO.

The position error in Figure 10 can be evaluated by the Root Mean Squared Error (RMSE) calculated by
(4)$$\mathrm{RMSE}=\sqrt{\frac{1}{n}\sum_{i=0}^{n}\left({\left(x_{i}-x_{i}^{truth}\right)}^{2}+{\left(y_{i}-y_{i}^{truth}\right)}^{2}\right)}.$$

The center coordinates of the tracking results and the ground truth in each frame are denoted by $\left(x_{i},y_{i}\right)$ and $\left(x_{i}^{truth},y_{i}^{truth}\right)$, where *n* is the number of frames. The center error and ROI error in the tracking results, the average of RMSE and IoU, are shown in Table 2.

According to Table 2, the proposed MAFiD method had a lower RMSE and a higher average IoU than MOSSE and YOLO alone. In terms of the frame rate, the output speeds of the MAFiD method and the control method were as fast as that of the MOSSE tracker because they used a tracker and a detector in two separate threads in parallel. However, the speed of thread 2 running in their backgrounds was different. In thread 2, the detection process in Figure 3 was performed. Thus, we can discuss the computational processing cost from Table 2. The processing speed of the MOSSE filter was 618 FPS and that of the YOLO detection was 70 FPS. When these methods were simply combined and run in parallel (control method), the processing time for detection was even slower, at 65 FPS with thread 2. In contrast, the MAFiD method can improve the detection cost to 95 FPS through more efficient processing. Since the model size and search area of YOLO was not the same for the control and MAFiD method, the processing cost caused a difference in thread speed. As a result, the MAFiD method could recover tracking early after the MOSSE tracking failure and obtain higher RMSE and IoU scores.

### 4.4. Detection Experiment

This experiment evaluated the detection performance. While the previous experiment measured inter-frame tracking performance, this experiment focused on the detection process from finding an object to starting the tracking process. In general, object detection tends to take a longer processing time than inter-frame tracking. Hence, we could also measure the bottleneck of the whole process through this experiment.

The methods to be compared in this experiment are listed below.

Proposed MAFiD MethodMOSSE tracker + YOLOv5 detector + Motion detectorDL-based detection methodYOLOv5 detectorROI-based TBD methodMOSSE tracker + YOLOv5 detectorControl methodMOSSE tracker + YOLOv5 detector (Global detection)

As well as DL-based detection, we need to compare hybrid methods that combine object tracking and object detection. In this experiment, we chose the TBD method, a conventional hybrid method, as a comparison. The TBD and MAFiD method both combine MOSSE and YOLOv5. The MAFiD method uses a correlation filter to switch between tracking and detection, whereas TBD uses the ROI directly as shown in Figure 4. In addition, to demonstrate the effectiveness of the proposed detection algorithm using a motion detector, we also compared the MAFiD method with the control method. The control method does not use motion detection but uses YOLOv5 to detect the entire image instead.

The experiments were conducted under two conditions: one in which the object moved at a relatively low speed and the other at a high speed. This allowed us to examine how differences in object velocity affected the tracking accuracy of each method.

#### 4.4.1. Slow Object Detection

The tracking results for each method when the object moved slowly are shown in Figure 12.

The velocity of the object was 1.1 m/s. Figure 12 shows how each method tracked the object’s position from the moment the object appeared in the camera frame. For the YOLO-only method and TBD tracking, it can be seen that the ROI position was shifted from the object position. This is attributed to the processing delay of YOLO. The TBD method updated the MOSSE tracker even when the ROI position was wrong. In contrast, the MAFiD method updated the tracker via a correlation filter, making it possible to output the object region accurately.

The figure below plots the center coordinates of object positions and IoU values.

Figure 13a shows that YOLO, TBD, and the control method detected the object slightly faster than the MAFiD method when it appeared in the frame. This was due to the difference in detection methods. YOLO can detect objects with high accuracy even if only a part of the object is visible. Therefore, conventional and control methods can start tracking at a relatively early stage. On the other hand, since the MAFiD method uses background subtraction with motion detection, the area of a moving object must be large in the subtracted image for detection. This caused the difference in detection delay between the MAFiD method and conventional methods.

In Figure 13b, the IoU values of YOLO and TBD are lower than those of the MAFiD method and control methods. This is because of the processing time delay. However, TBD did not show oscillations in IoU values like YOLO. TBD could update the ROI with the MOSSE tracker even during YOLO detection, thus reducing the position error of TBD over YOLO.

The RMSE and average IoU values for each method are shown in Table 3.

In Table 3, the proposed MAFiD method had a larger RMSE error than the TBD and control methods. This result was caused by the difference in how quickly they detected slow-moving objects, as shown in Figure 13a. Meanwhile, in terms of the average IoU value, the MAFiD method scored higher than the conventional method. This indicates that the MAFiD method could correct the errors after detection and continue the tracking process with higher accuracy, whereas the conventional methods had errors due to YOLO delays.

#### 4.4.2. Fast Object Detection

The same experiment was performed under conditions where the object was moving fast. Figure 14 shows tracking using each method in a fast object video.

The velocity of the object was 3.7 m/s. In Figure 14, when an object appeared in the frame, the MAFiD method updated the ROI in 5.6 ms. Since the MAFiD method updated only the center position of the ROI by motion detection, the size and aspect ratio were not accurate at this stage. Despite this drawback, it has the advantage of starting CF-based tracking earlier than other methods. After finding the object, the size and aspect ratio of the ROI were corrected in 30.9 ms while tracking the center position. On the other hand, the other methods took time to update the ROI, resulting in ROI errors when capturing high-speed subjects. Compared to Figure 12, it can be seen that the error became larger as the object moved faster. The TBD and control methods started tracking after the YOLO output detection results. Thus, if the detection position is shifted as in this experiment, subsequent tracking will also fail. Consequently, the performance bottlenecks of the TBD and control methods were equal to or less than that of the YOLO-only method, indicating that the MOSSE tracking process did not contribute enough.

The following is a plot of the coordinates of the object’s center location and IoU.

From the x-coordinate in Figure 15a, the proposed MAFiD method could track the actual object position after detection, whereas the conventional method showed a tracking delay. In particular, it can be noted that there were delays not only in YOLO and TBD but also in the control method. Moreover, the y-coordinate graph shows that the MAFiD method located the ROI the fastest, while the control method lagged far behind. This suggests that the MAFiD method can start tracking earlier than the conventional method by updating the correlation filter through fast motion detection. On the other hand, the control method took a longer processing time than the conventional method since the tracker started running after objects were detected by YOLO. In the case of a high-speed object, as in this experiment, the position error was too large to be corrected by the MOSSE filter. Therefore, the object did not exist in the search range of the correlation filter at the detection-to-track phase. In summary, by comparing the control method and the MAFiD method in Figure 13a and Figure 15a, it can be shown that motion detection contributed to detecting fast objects.

The IoU graph in Figure 15b shows that there was a significant difference in accuracy between the MAFiD method and the other methods. This is attributed to the slow detection by YOLO and the failure of MOSSE tracking. In addition, unlike the other methods, the IoU score of the MAFiD method increased to nearly 1.0 at around 40 ms. This indicates that the MAFiD method successfully matched the ROI by reflecting the detection results while performing fast tracking in parallel processing.

The RMSE and average IoU values for each method are shown in Table 4.

Table 4 shows that the proposed MAFiD method had relatively high accuracy in both RMSE and average IoU. In particular, the difference in IoU between the MAFiD method and the conventional method was large, meaning that the MAFiD method was superior in fast object tracking. Furthermore, the accuracy difference between the control method and the MAFiD method was also confirmed. It can be said that low-cost object detectors, such as motion detection, play an important role in high-speed object tracking.

#### 4.4.3. Detection Latency

Based on the two detection experiments in this section, the latency obtained by measuring the processing time for each method is shown in Table 5.

Since the YOLO method was detected frame by frame, the detection latency in Table 5 was the same as the tracking frame rate in Table 2. The TBD and control methods required both YOLO execution time to detect the entire frame image and MOSSE filter update time, which made the detection time relatively slow. On the other hand, the proposed MAFiD method was based on faster motion detection, enabling it to track the object’s location faster than other methods. In addition, the MAFiD method had a shorter latency than the other methods, including the detection time to update the size and aspect ratio of the ROI. This is due to the parallel processing of multiple methods and the use of local detection by YOLO. The local detection of the MAFiD method searched only around the ROI output by the tracker, resulting in reduced processing costs for ROI detection. From the above, it can be concluded that the MAFiD method had less delay than the conventional method and the control method in Table 5.

## 5. Conclusions

In this study, we proposed a method for tracking moving objects with high speed and high accuracy. The proposed algorithm combines correlation filter-based tracking, deep learning-based detection, and motion detection using background subtraction to improve both tracking and detection performance. As for the tracking process, we propose a mechanism to perform MOSSE tracking and YOLOv5 detection in parallel while assisting each other’s processing. The experiment achieved a frame rate of 618 FPS and an IoU accuracy of 84%. Regarding the detection process, the proposed MAFiD method focuses on dealing with a fast-moving object. By using motion detection, it can quickly locate object positions and regions. Experimental results show that the MAFiD method was more accurate and faster than conventional detectors, especially for objects moving at high speed. The delay was found to be 3.48 ms for position detection and 11.94 ms for ROI detection. In summary, the MAFiD method proposed in this work will contribute to the improvement of object-tracking technology, both in terms of fast object tracking and detection.

This study deals with single-object-tracking methods. In practical situations, however, there are many cases where the target object must be handled among several other objects. Therefore, the next task for the practical application of this study is to apply object recognition to the proposed MAFiD system. To achieve this, it would be effective to train YOLO with multiple types of objects for detection so that it can selectively track only certain targets. Future improvements are also needed to deal with occlusion and to enable tracking in complex environments with dense objects.

## Figures and Tables

**Figure 1 sensors-23-07082-f001:**
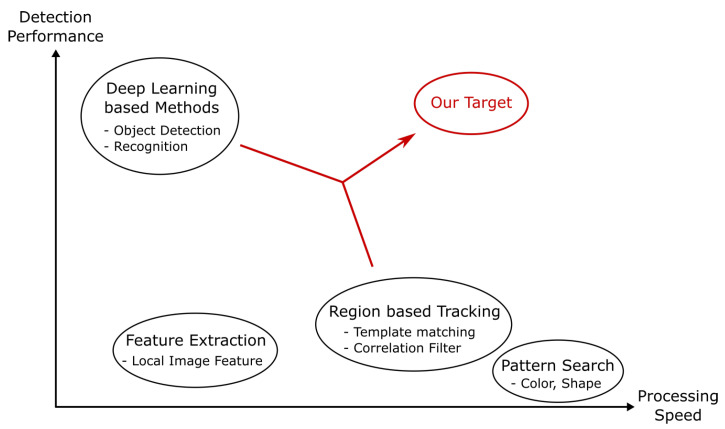
Motivation for this study.

**Figure 2 sensors-23-07082-f002:**
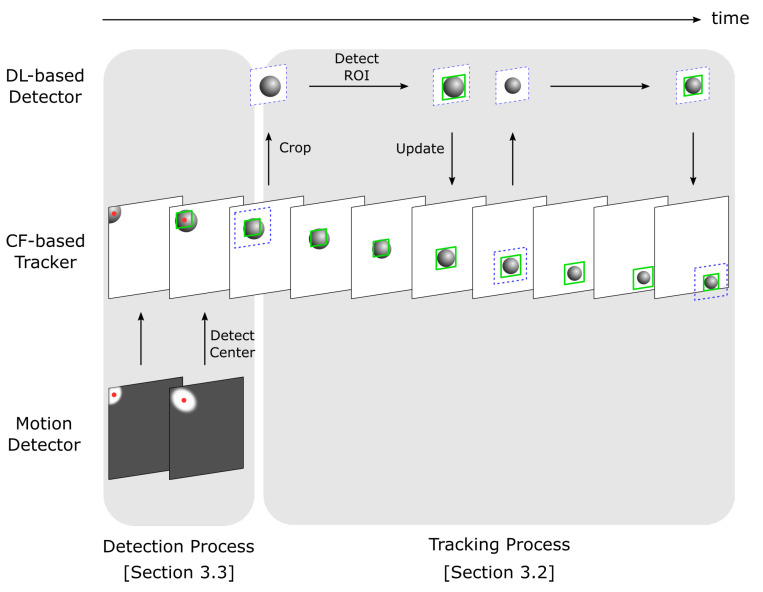
Concept of the proposed MAFiD method.

**Figure 3 sensors-23-07082-f003:**
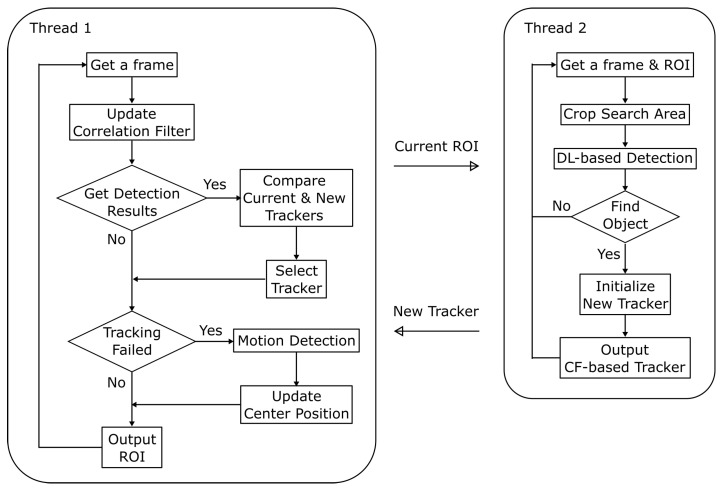
The overall process flow of the proposed MAFiD method.

**Figure 4 sensors-23-07082-f004:**
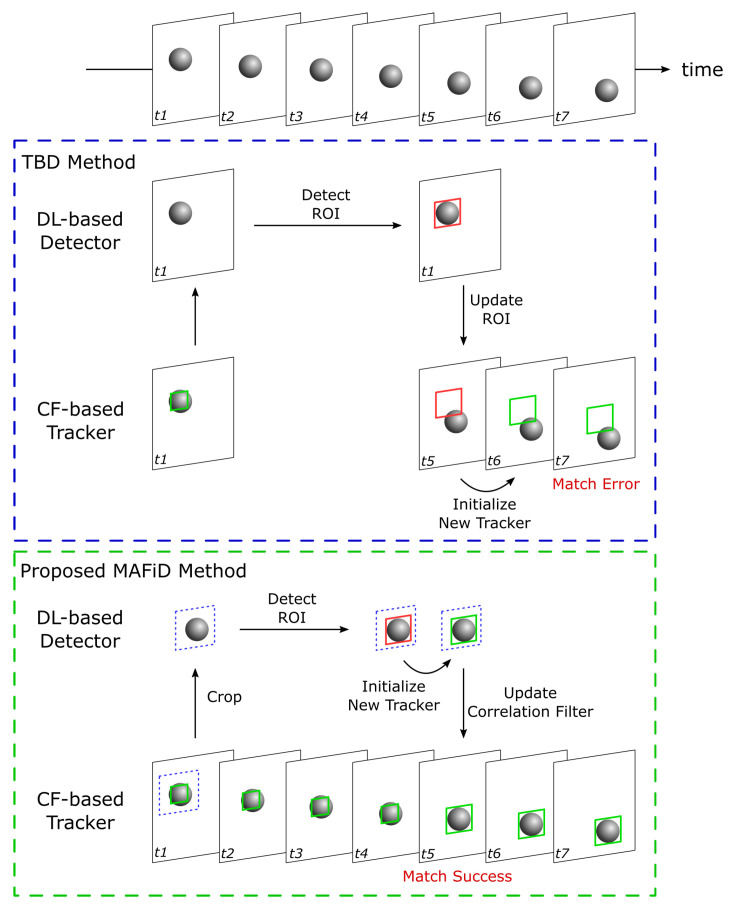
Comparison of tracking processes in TBD and the proposed MAFiD method.

**Figure 5 sensors-23-07082-f005:**
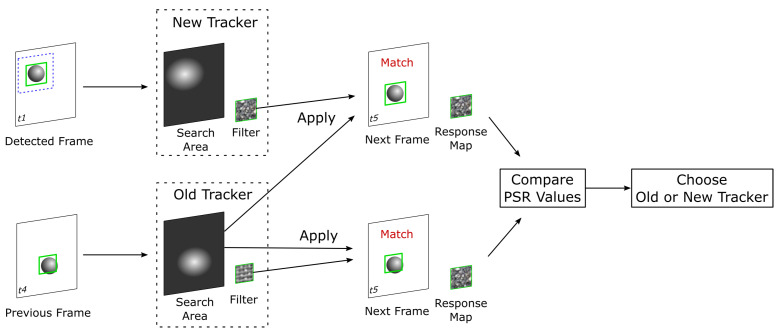
Filter update method in the proposed MAFiD method.

**Figure 6 sensors-23-07082-f006:**
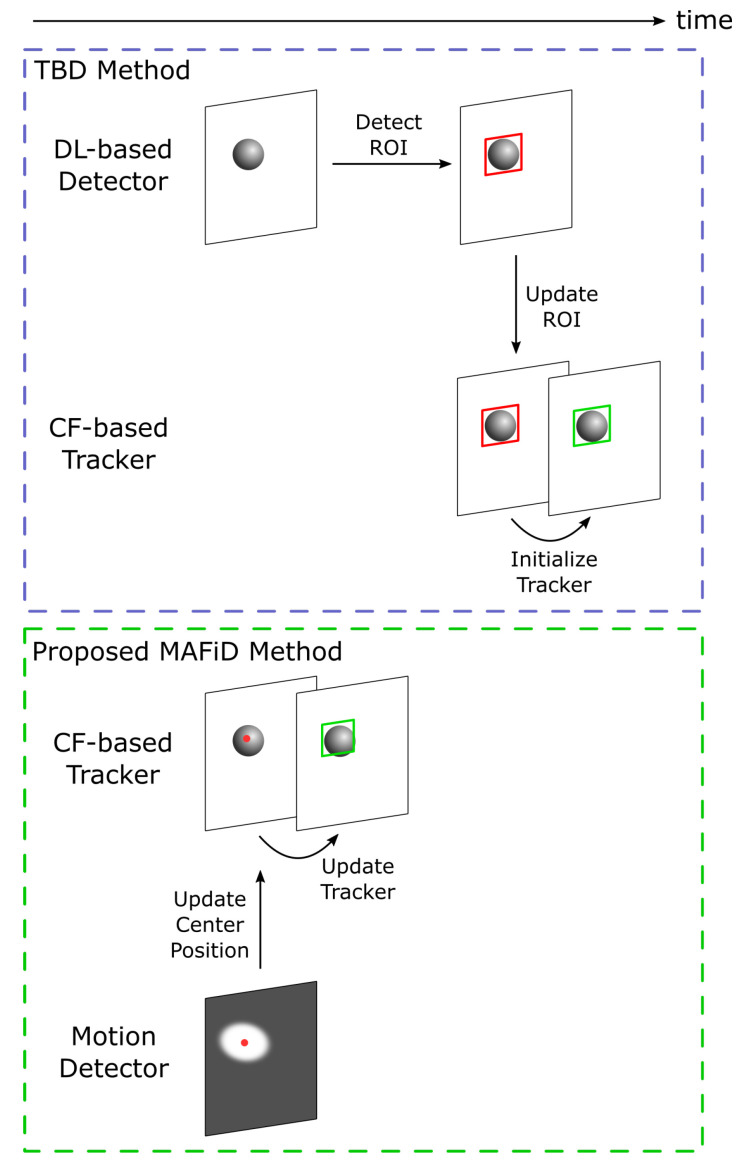
Comparison of detection processes in TBD and the proposed MAFiD method.

**Figure 7 sensors-23-07082-f007:**
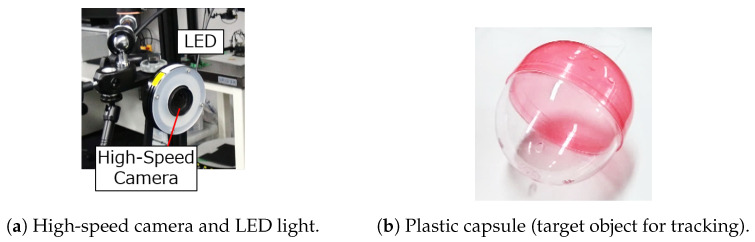
Experimental devices.

**Figure 8 sensors-23-07082-f008:**
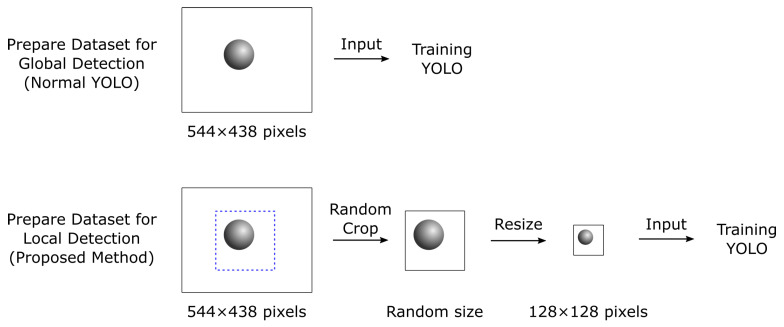
Preparation of training data.

**Figure 9 sensors-23-07082-f009:**
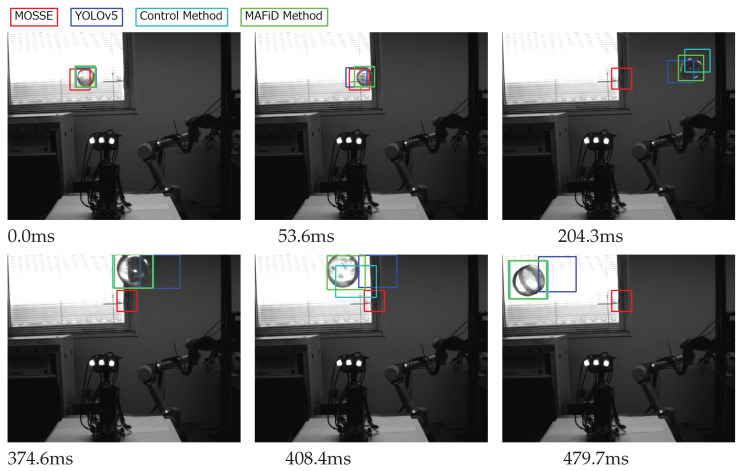
Comparison of tracking by each method.

**Figure 10 sensors-23-07082-f010:**
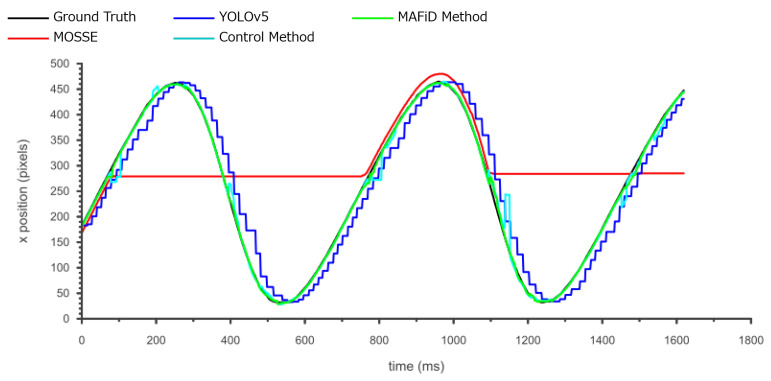
X-coordinate of the tracking position for each method.

**Figure 11 sensors-23-07082-f011:**
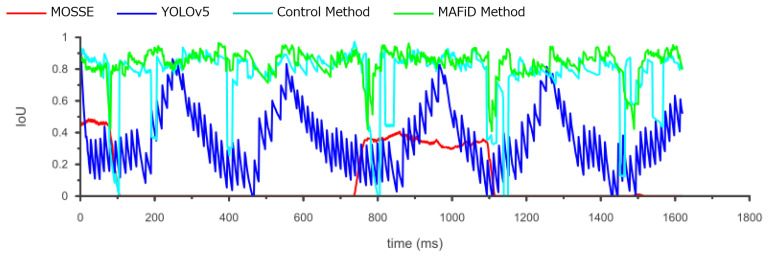
Change in the IoU for each method.

**Figure 12 sensors-23-07082-f012:**
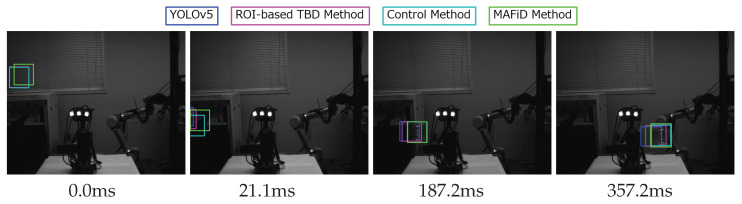
Detection process for each method when an object moved slowly (1.1 m/s).

**Figure 13 sensors-23-07082-f013:**
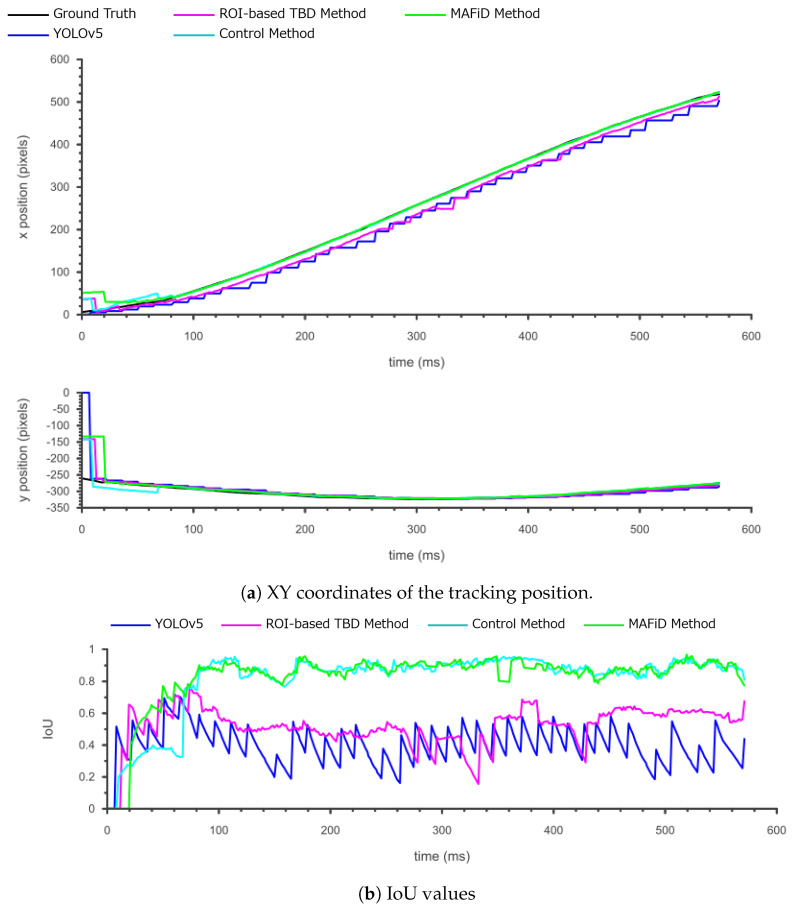
Tracking results for each method when the object moved slowly (1.1 m/s).

**Figure 14 sensors-23-07082-f014:**
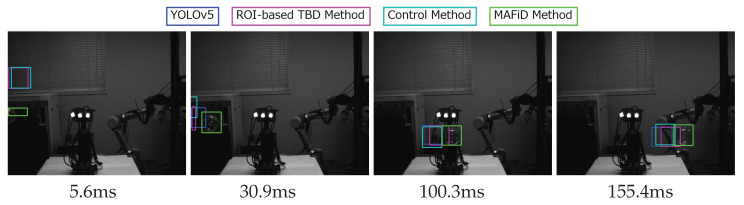
Detection process for each method when an object moved fast (3.7 m/s).

**Figure 15 sensors-23-07082-f015:**
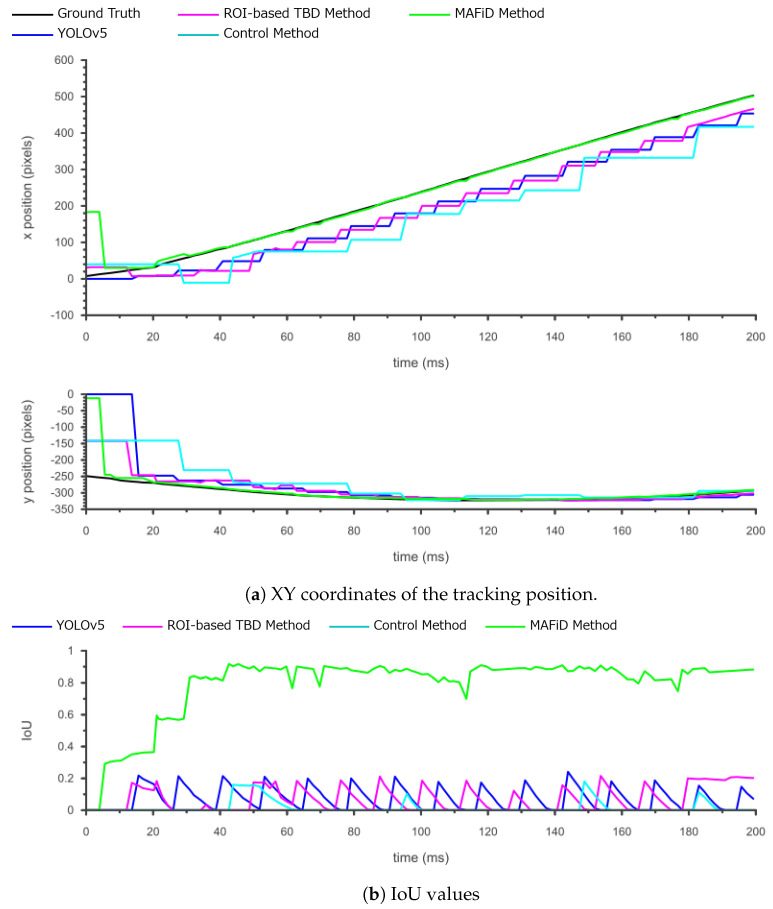
Tracking results for each method when the object moved fast (3.7 m/s).

**Table 1 sensors-23-07082-t001:** YOLO training results on each of the two datasets.

Image Size of Dataset	mAP (0.5)	mAP (0.5–0.95)
544 × 438	0.99	0.89
128 × 128	0.99	0.86

**Table 2 sensors-23-07082-t002:** Tracking performance of each method.

Method	FPS	RMSE (Pixels)	Average IoU
MOSSE tracker	618	141.0	0.10
YOLOv5 detector	70	35.1	0.36
Control method	618 (Thread 2: 65) ^1^	14.5	0.76
MAFiD method	618 (Thread 2: 98) ^1^	4.9	0.84

^1^ Speed of the loop processing in thread 2 for object detection in parallel processing, as shown in Figure 3.

**Table 3 sensors-23-07082-t003:** Detection performance of each method when the object moved slowly (1.1 m/s).

Method	RMSE (Pixels)	Average IoU
YOLOv5 detector	39.2	0.40
TBD method	25.8	0.52
Control method	17.9	0.81
MAFiD method	27.3	0.83

**Table 4 sensors-23-07082-t004:** Detection performance of each method when the object moved fast (3.7 m/s).

Method	RMSE (Pixels)	Average IoU
YOLOv5 detector	89.1	0.07
TBD method	60.6	0.07
Control method	92.0	0.01
MAFiD method	53.1	0.78

**Table 5 sensors-23-07082-t005:** Latency of detection process for proposed MAFiD method and conventional methods.

Method	Detection Latency (ms)
YOLOv5 detector	14.3
TBD method	15.4
Control method	15.4
MAFiD method	3.48 (ROI: 11.94) ^1^

^1^ Latency to detect the ROI size and aspect ratio, which are larger than the latency to find the center position.

## Data Availability

Not applicable.

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
