# Peer review of "High-Speed Tracking with Mutual Assistance of Feature Filters and Detectors"

_sensors, 2023, doi:10.3390/s23167082_

Round 1

Reviewer 1 Report

In  this study, the authors proposed a method for tracking moving objects with high speed  and high accuracy. The proposed algorithm combines correlation filter-based tracking deep learning-based detection, and motion detection using background subtraction to improve both tracking and detection performance. However, minor corrections are required as follows:

- As the proposed approach combines three methods, it is required to discuss the complexity of the proposed system/solution in terms of efficiency/computational time. This is important because it deals with real-time high speed objects. 

- it us required to describe the two datasets well. 

- it is mentioned that (YOLOv5 wastrained on 8,000 images as training d ata and the remaining 800 images 434 as cross-validation data). It is not clear: did you use cross validation for trainings/testing? If so, how many folds you used. If not, revise this section and explain the split ratio for training, testing & validation. 

- The evaluation metrics need to be described well in the methodology.

It is well written.

Author Response

We appreciate your valuable review comments. We have uploaded our response to your comments in a Word file.

Reviewer 2 Report

I appreciate the work of A Tracking approach that incorporates several techniques:

Deep learning-based object detection, motion detection, and correlation filter-based object tracking with backdrop removal. The algorithms operate concurrently and aid one another in processing. 

However there few things which are not answered over the paper.

·         What are the limitations or restrictive assumptions behind the proposal?

·         Introduction and survey can be done extensively with the latest papers also such as PP YOLO, YOLOX…

·         How do you overcome the drifting problem in MOSSE?

·         Since YOLOV5 has difficulty in detecting smaller groups how does that support the real time applications

·         The legend with proposed can be renamed as MOSSE tracker + YOLOv5 detector + Motion detector

·         The contents are repetitive in Tracking Experiment 4.3 and Detection Experiment 4.4. Can be organized in a unique way.

·         What dataset was used and also the dataset size can be mentioned.

Author Response

(The authors gave the same response as above.)

Reviewer 3 Report

Aiming at the difficulty of increasing tracking speed and detection accuracy simultaneously in camera images, the paper analyzes the advantages and disadvantages of the conventional methods, then a tracking method is proposed, which combines correlation filter-based tracking, deep learning-based detection, and motion detection using background subtraction. The method optimizes both the tracking process by filter updating and the object detection process by introducing motion detection. Experimental results show that the proposed tracker improved the overall performance of system and was more accurate and faster than conventional detectors, especially for objects moving at high speed.

Improvement suggestions,

1.       The title may be improved make it more consist with the content.

2.       The proposed hybrid method may need to be given a specific name to facilitate the discussion in this paper and to be cited conveniently by others.

3.       The caption of figure 3 may be improved, it looks like the figure should be cited in Section 3.2 which is named as tracking process.

4.       The arrow in Figure 4 indicates that the proposed method only uses detected ROI in the step of initializing new tracker, which is the same as TBD and not consistent with the description in line 293 .

Author Response

(The authors gave the same response as above.)

Reviewer 4 Report

To increase tracking speed and detecting accuracy for the moving object, some methods of a correlation filter-based object tracking, deep learning-based object detection, and motion detection with background subtraction are combined in this manuscript. Those algorithms work in parallel and assist each other. The proposed processing can improve the overall performance of the system. This work may be significant in application area. However, there are a lot of inappropriate writing expressions that need to be improved.

For example,

Fig 1 is unprofessional, the elements of the diagram should be more closely aligned.

Fig 2 is unprofessional, tracking process in Section 3.2 is absent in the third line.

Fig 13(a) and Fig 15(a) must be expressed in 3-dimensional style.

Author Response

(The authors gave the same response as above.)

Round 2

Reviewer 1 Report

The paper is revised well and considered all the comments given. Thank you. 

A minor editing is required to remove any potential typos. 

Reviewer 4 Report

In revised manuscript, a method for tracking moving objects with high speed and high accuracy was discussed. A correlation filter-based tracking, deep learning-based detection, and motion detection using background subtraction have been combined to improve both tracking and detection performance. A mechanism to perform the MOSSE tracking and the YOLOv5 detection have been presented in parallel processing. The experiment achieved, such as regarding the detection process and dealing with a fast-moving object. The conclusion shows significantly. All comments have been revised. Therefore, this reviewer thinks the manuscript can be accepted for publication.